# Identification of microRNA Signatures in Peripheral Blood of Young Women as Potential Biomarkers for Metal Allergy

**DOI:** 10.3390/biomedicines11020277

**Published:** 2023-01-19

**Authors:** Zhang Yuehui, Maki Hosoki, Masamitsu Oshima, Toyoko Tajima, Mayu Miyagi, Swarnalakshmi Raman, Resmi Raju, Yoshizo Matsuka

**Affiliations:** 1Department of Stomatognathic Function and Occlusal Reconstruction, Graduate School of Biomedical Sciences, Tokushima University, Tokushima 7708504, Japan; 2Eunice Kennedy Shriver National Institute of Child Health and Human Development (NICHD), National Institutes of Health (NIH), Bethesda, MD 20892, USA

**Keywords:** allergic contact dermatitis, biomarker, metal allergy, microRNA, peripheral blood

## Abstract

MicroRNA (miRNA) is a short (19–24 nucleotide) endogenous non-protein RNA that exists in the body and controls the translation process from genes to proteins. It has become useful as a diagnostic tool and a potential treatment target in cancer research. To explore the function of miRNA in contact dermatitis, female participants with a positive metal allergy diagnosis (*n* = 3) were enrolled along with additional female participants with no medical history of metal allergy (*n* = 3). A patch test was performed on each participant. Peripheral blood was collected from all the participants before the patch test and at days 3 and 7 after starting the patch test. After total RNA was obtained from peripheral blood leukocytes and cDNA was generated, microarray analysis was performed to analyze the large-scale circulating miRNA profile. Real-time polymerase chain reaction (RT-PCR) was then used to clarify the overall target miRNA expression. Downregulation of hsa-let-7d-5p, hsa-miR-24-3p, hsa-miR-23b-3p, hsa-miR-26b-5p, and hsa-miR-150-5p was found on day 7. Certain miRNAs were confirmed using RT-PCR. These peripheral blood miRNAs could be diagnostic biomarkers for metal allergies.

## 1. Introduction

Metals such as gold (Au), silver (Ag), platinum (Pt), palladium (Pd), nickel (Ni), chromium (Cr), cobalt (Co), titanium (Ti), and mercury (Hg) are widely used in numerous applications. Metal alloys, specifically, are commonly used in items such as apparel accessories, coins, mobile phones, and dental materials, and people often come into contact, by touch, with everyday items containing metal alloys [1]. Allergic contact dermatitis (ACD) is typically caused by exposure of the skin to allergens, such as metal products, chemicals, and green plants [2]. Metal allergy is an inflammatory illness that is classified as a delayed-type hypersensitivity (DTH). Approximately 10–15% of the general population has symptoms of contact allergy to metal alloys [1,3]. This type of skin allergy is more common among women than men, with a population frequency of about 10% for women and 2% for men [4]. Additionally, because younger people often wear earrings and other accessories, they have a higher incidence of metal allergies. Metal allergies have increased significantly in Japan [5,6]. In clinical medicine, metal allergy is related to the causes of allergic dermatitis, palmoplantar pustulosis, lichen planus, sweating skin eczema, and burning mouth syndrome, which threaten people’s physical and mental health [7,8].

Patch testing is the first choice for diagnosing metal allergies. However, it is a relatively objective test with some subjective factors. A diagnosis that is made using patch testing is based on inspection and palpation of the testing area. There is a risk of misdiagnosis because the results can be misread [9]. Other disadvantages include a false positive reaction due to the process of reagent application and the necessity of multiple visits before a final diagnosis can be obtained [10]. The patch test determines sensitization of the causative agent but does not determine the severity of symptoms. It will be beneficial in clinical practice if we can establish a biomarker that can evaluate symptoms and severity, as it could be utilized for the prevention of disease.

ACD that is caused by metallic materials is considered to be an inflammatory disease. Although sensitization to Ni is the main cause of ACD induced by metallic materials, sensitization to Co, Cr, Pd, and Au involves immune responses, which is similar to the previously demonstrated production of mixed Th1 and Th2 cytokines that are induced by Ni [11,12]. Generally, ACD that is caused by metallic materials is thought to involve the innate and adaptive immune systems. Within the overall process of metal-induced ACD, different complex pathways and overlapped networks are activated [13,14,15,16,17]. Although scientific research continues to develop, the sensitization phase is extensively characterized and immunological events take place during the elicitation phase, but these events have not been completely elucidated [18].

MicroRNA (miRNA) is a short (19–24 nucleotide) endogenous non-protein RNA that can negatively control gene expression through the non-specific base pairing of the overall target mRNA coding sequence, which in turn, causes transcript cleavage or translational repression. Over 60% of human protein-coding genes contain miRNA target sites [19]. miRNA may be able to adjust some genes. Because each miRNA functionally targets several (rather than individual) genes, they are of great interest from a therapeutic perspective [20]. Over the past few decades, miRNAs have undergone in-depth analysis as biomarkers for cancer diagnosis, prognosis, and treatment [21]. Additionally, the potential to modify miRNA responses in inflammatory skin diseases such as psoriasis, allergic asthma, food allergy, atopic dermatitis, and ACD is gradually being revealed [22,23,24].

Although there are many scientific studies on the participation of miRNAs in inflammatory skin diseases (such as atopic dermatitis), previous discussions on miRNA efficacy in ACD inflammation are limited [25,26,27,28,29]. Vennegaard et al. reported the first scientific study on miR-21, miR-142-3p, miR-142-5p, and miR-223 upregulation in a diphenylcyclopropenone (DPCP)-challenged skin biopsy from humans and mice using microarrays [26]. Then, using deep sequencing, a comprehensive miRNA expression profile that was based on the DPCP-challenged skin biopsy at three different time points was created [27]. Recently, a miRNA–mRNA profile in the skin of patients who were exposed to allergens (Ni, epoxy resin, and hydroxychloroisothiazolinone) was reported [28]. Additionally, skin biopsy is more commonly used in miRNA research related to ACD [29], but there have been few studies investigating the role of blood cell miRNA in ACD responses.

Herein, to decipher the peripheral blood cell involvement and the miRNA-related immunological pathways in more detail, we analyzed miRNA expression profiles in peripheral blood leukocytes from both non-allergic and allergic young women who were exposed to multiple metals.

## 2. Materials and Methods

### 2.1. Study Participants and Patch Test

The subjects were carefully selected because comprehensive analysis was difficult in this study. Because of the high frequency of female patients with ACD, female subjects were selected for this study. Healthy volunteers were candidates who had no underlying disease or history, especially of other allergic diseases (allergic rhinitis, atopic dermatitis, food allergy, etc.), no dental restorations in the oral cavity, and no history of orthodontic treatment. Young women were selected as candidates for this study, because there were few subjects who met these criteria as they got older. Subjects with a diagnosis of metal allergy were candidates who were in the same age range as the healthy volunteers, had no underlying medical conditions, had or had had metal contact dermatitis, were not receiving any medication, and had no allergic rhinitis or asthma. Candidate subjects underwent patch testing and blood sampling. In the healthy volunteer group, subjects were excluded from the study if they had any suspicious reactions to the patch test on any day of the study. In the patient group, subjects were included if they had an obvious + reaction or an elevated response on day 7 compared to day 3. Those subjects were excluded if the intensity of reaction on day 7 was reduced or disappeared compared to day 3. Six young female participants (mean age, 24 years; range, 22–28 years) were recruited. The patch test protocol was conducted in accordance with the standards of the Metal Allergy Clinic in the Stomatology department at Tokushima University Affiliated Hospital.

The study protocol was conducted in accordance with the Declaration of Helsinki and was approved by the Research Ethics Committee at Tokushima University Hospital (number 1036).

### 2.2. Patch Test Procedure

All participants underwent a general examination and an intra-oral examination by dentists with more than 20 years of experience in the Dental Metal Allergy Clinic. A patch test with 28 types of metal allergens was then performed to identify allergenic metallic material.

Our clinic uses 19 patch test metal allergens (17 from Patch Test Reagents; Torii Pharmaceutical Corp., Tokyo, Japan and two from allergEAZE; SmartPractice Canada Corp., Calgary, AB, Canada) and nine custom-made allergens [30,31]. These allergens include most of the metallic elements that are the main ingredients in alloys used for dental treatment in Japan. These allergens were attached to the skin on the back of the subjects using an adhesive bandage (Patch Tester Torii; Torii Pharmaceutical Corp., Tokyo, Japan) and removed after 2 days. Skin reactions were classified at 2, 3 and 7 days after placing the adhesive bandage using the criteria that were determined by the International Contact Dermatitis Research Group [10,31,32,33].

### 2.3. Blood Samples and RNA Extraction

Peripheral blood was collected from all participants before the patch test, and 3 and 7 days after starting the patch test. All the blood samples were preserved in 5 mL blood tubes (BD Medical 367,886 Vacutainer, Kernersville, NC, USA). Total leukocyte RNA was isolated from the peripheral blood sample using the QIAamp RNA Blood Mini Kit (QIAGEN, Germantown, MD, USA), in accordance with the manufacturer’s instructions.

### 2.4. Microarray Analysis

The quality and quantity of total RNA were obtained using photometry (Nanodrop ND-1000, NanoDrop Technologies, Inc., Wilmington, DE, USA) and an Agilent Bioanalyzer 2100 (Agilent Technologies, Santa Clara, CA, USA). The obtained miRNA was analyzed using a miRNA array chip (3D-Gene, TORAY, Tokyo, Japan). These data were obtained in cooperation with TORAY Industries, Inc. Fluorescent signals that were obtained on the chip were analyzed using 3D-Gene Extraction software (Toray Industries Inc., Tokyo, Japan). The microarray data are in agreement with the Minimum Information About a Microarray Experiment (MIAME) and are publicly available through the Gene Expression Omnibus (GEO) database and GeneSpring (http://www.ncbi.nlm.nih.gov/projects/geo/ (accessed on 13 August 2019)).

### 2.5. Processing Microarray Data

For each miRNA probe, the compressive strength of most probes was measured on the basis of the mean scaling value that was measured from each group by calculating a modified t-statistic called the significance analysis of microarrays [24]. The adjusted t-statistic was used to evaluate the different expression levels in the groups. Briefly, we calculated the mean log two-fold change of each miRNA for all six subjects at three different time points. Using novel self-amplifying mRNA for each miRNA, we evaluated whether the average log two-fold change was significantly different from zero, and sorted all of the miRNAs from most to least significant. An ingenuity pathway analysis, Gene Ontology Analysis, and KEGG pathway enrichment analysis were performed.

### 2.6. Real-Time Quantitative Polymerase Chain Reaction

The TaqMan MicroRNA Reverse Transcription Kit was used followed by an individual real-time quantitative polymerase chain reaction (RT-qPCR) test to verify the hsa-let-7d-5p, hsa-miR-24-3p, -23b-3p, -26b-5p, and -150-5p expression levels (Applied Biosystems, Carlsbad, CA, USA) using the TaqMan Advanced MicroRNA test (Applied Biosystems). In accordance with the manufacturer’s instructions, the StepOnePlus PCR system software (Applied Biosystems) was used to conduct the PCR test. The RT-qPCR data information was solved using the ΔΔCT calculation method to obtain the fold changes of the expression [34], and the hsa-miR-423-5p was used as the reference gene, in accordance with the guidelines.

### 2.7. Statistical Analysis

The Mann–Whitney U-test and a *t*-test with unpaired data information were used to compare differences between groups. The graph was obtained using GraphPad (GraphPad Software Inc, San Diego, CA, USA), and *p* < 0.05 was found to indicate a statistically significant difference.

## 3. Results

### 3.1. Results of Patch Test

All the participants were young women between 22 years and 28 years of age (Table 1). The participants in the control group did not have any sign of inflammation, and participants in the patient group showed signs of Ni–Co allergic dermatitis, ranging from mild macular erythema to a strong reaction (Figure 1, Table 2).

### 3.2. Microarray Analysis

After RNA extraction and quality control, a comprehensive miRNA profile was generated on the basis of the microarray analysis.

Globally, 571 miRNAs were expressed, and 63 miRNAs were differentially expressed (Figure 2a). A microarray significance analysis showed that there were significant differences in the expression of certain miRNAs among the different time points in the patient group. However, a significant difference was not found at any time point in the control group (Figure 2b). Although the visible signs peaked 3 days after starting the test in the patient group, the miRNAs showed no difference between day 0 and day 3 or at day 3 and day 7. There were 20 significantly decreased miRNAs at day 7 compared with day 0 (Figure 2c). This may indicate that although day 3 was considered to be the local peak reaction to the patch test, day 7 could be the key time point in the blood during an elicitation reaction. The miRNA signature from the peripheral blood leukocytes may represent a systemic component rather than local miRNA changes in the skin.

### 3.3. Pathway Analysis

Ingenuity pathway analysis was performed for microarray data at different time points in the control and patient groups. The deregulated miRNAs on day 7 in the patient group were mostly related to inflammatory disease and the inflammatory response (Figure 3). This further supported the possible relationship between deregulated miRNA and ACD reaction. Five deregulated miRNAs (hsa-let-7d-5p, hsa-miR-23b-3p, hsa-miR-24-3p, hsa-miR-26b-5p, and hsa-miR-150-5p) were selected out of 20 on the basis of Ingenuity Pathway Analysis miRbase (https://www.mirbase.org/ (accessed on 5 August 2019)), and TargetScanHuman 7.2 (http://www.targetscan.org/vert_72/ (accessed on 23 October 2019)) (Figure 4). For hsa-miR-24-3p and hsa-miR-150-5p, due to the small number of participants, the *p*-value for certain miRNAs was greater than 0.05. However, these two miRNAs showed a deregulation tendency at days 0, 3, and 7. Additionally, the TargetScanHuman 7.2 database revealed that hsa-miR-24-3p and hsa-miR-150-5p might have the best possibility to participate in Th1 and Th2 cell pathways. Therefore, these two miRNAs were identified as target genes (Figure 5).

### 3.4. Real-Time qPCR Results

Validation by RT-qPCR was performed using TaqMan advanced assays to examine only the mature forms of certain miRNAs. The detailed primer design is shown in Table 3. Target gene expression was normalized using hsa-miR-423-5p expression. hsa-miR-423-5p was chosen as the reference because its expression in the microarray is more stable than most common endogenous molecules, in contrast to U6. Hsa-let-7d-5p, hsa-miR-23b-3p, and hsa-miR-150-5p were shown to be significantly deregulated on day 7 compared with days 0 and 3 in the patient group, hsa-miR-24-3p and hsa-miR-26b-5p were shown to be significantly deregulated on day 7 compared with day 0 (Figure 6).

GO analysis identified the top 12 most related functions for the miRNAs that showed changes. KEGG pathway analysis revealed 18 pathways in which the miRNAs that showed change could participate (*p* < 0.05) (Figure 7).

## 4. Discussion

In this study, we analyzed miRNA expression patterns in peripheral blood leukocytes from control group participants who showed negative patch test results and from patient group participants who showed a partial positive reaction to the patch test. We found three significantly downregulated miRNAs (hsa-let-7d-5p, hsa-miR-23b-3p, and hsa-miR-150-5p) at day 7. Furthermore, two miRNAs with an obvious deregulating tendency at day 7 were also detected. Downregulation of these five miRNAs was confirmed by qPCR.

To the best of our knowledge, this is the first report to investigate miRNA expression in peripheral blood leukocytes in a human ACD reaction using metallic materials as sensitizers at different time points. ACD is generally described as a T cell-mediated disease that is mainly mediated by cytotoxic CD8 T cells and Th1 cells [35]. The five miRNAs that showed the greatest tendency for downregulation among all the expressed miRNAs were shown to be associated with inflammatory diseases or inflammatory responses, especially those associated with the activation of T cells and T cell-related pathways, which is consistent with the fact that allergic contact dermatitis is mainly a T cell-mediated skin disease [36]. To date, none of these five miRNAs has been reported to be directly related to human ACD sensitizing metal ions. This study can be considered as a pioneering study in this respect.

To date, only one report has shown the miRNA profile in skin and peripheral blood mononuclear cells (PBMCs) during Ni allergy in humans [29]. This report indicated that there were 19 upregulated miRNAs in PBMCs 3 days after Ni ion sensitization. Our results showed a different release profile than those in this report because this report examined miRNA expression in skin and PBMCs. Our research used the whole blood buffy coat to obtain leukocytes instead of isolated PBMCs because we replicated the most easily organized samples in clinical trials. The changes in miRNA expression in whole blood leukocytes might be considered to be a systematic regulator for overall immunomodulation during the sensitization and elicitation of a reaction by allergens. Subjects with a common positive reaction to Ni and Co were selected because these two metal ions most frequently cause metal allergies among humans [37,38]. In this study, the patch test applied not only Ni but also multiple reagents, and targeted subjects who were positive for several reagents. This is because multiple metal reagents are applied in clinical practice, and we observed the profile of miRNA as a comprehensive systemic reaction. Although we showed the miRNA profile at three different time points during the patch test process, our data only showed a substantial miRNA downregulation at day 7. In future experiments, it was suggested that effective changes could be obtained by only comparing day 0 and day 7. In addition, on day 0, the miRNA profile tended to be higher in the patient group than in the control group, but the difference was not significant. If the difference on day 0 becomes clear, it may be possible to use it as a differential diagnosis tool for ACD, so further investigation was considered necessary. The five miRNA profiles we identified have been shown to be unique and not just non-empty subsets of each other; they may have a positive effect against the negative regulation of immune responses in the ACD process in humans. For example, miR-150 showed slight downregulation on day 3, but the tendency toward deregulation became obvious at day 7. This preferential miR-150 expression during the patch test suggests that this miRNA inhibits the DTH response in mice [39]. Thus, it may be necessary to deregulate early in a DTH reaction to resolve the inflammation. All five miRNAs that were downregulated at day 7 are related to cancer and play a role in reducing cell proliferation [40,41,42,43,44]. miR-24-5p, miR-23b-3p and miR-26b-5p upregulation has been reported in atopic dermatitis and psoriasis. Additionally, let-7d was upregulated in asthma [45,46,47]. miR-150 has been reported to be downregulated in psoriatic skin lesions [48]. Thus, the five microRNAs that we found to be deregulated in ACD are also involved in other skin inflammatory diseases, which may not be surprising because T cells play an important role in the pathogenesis of all three diseases. However, the role of the altered miRNA profile in ACD, as well as in psoriasis, atopic dermatitis, and asthma, in humans is largely unknown. The miRNA profile in inflammatory disease is slowly being pieced together by exploring the specific pathways and target mRNAs.

Using TargetScanHuman 7.2 database, we found that these five downregulated miRNAs were related to Th1 cell differentiation, Th2 cell differentiation, and apoptosis activity (Figure 5), which could affect the ACD reaction on three different levels. Thus, miR-24-3p, miR-26b-5p, and miR-150-5p were shown to have a direct relationship with the following proinflammatory cytokines: interleukin (IL)-4, IL-5, IL-13, and interferon (IFN)-γ. IL-10 and IFN-γ were reported to be significantly upregulated in human PBMCs when sensitized with Ni as an allergen [49]. We speculate that the decrease in let-7d-5p, miR-23b-3p, miR-24-3p, miR-26b-5p, and miR-150-5p expression at day 7 of the patch test was due to proinflammatory cytokine upregulation. Further studies are required using the metal allergy model to confirm the miRNA–mRNA interaction. Another possible mechanism of T cell infiltration from blood vessels into the dermis and epidermis could be due to the miRNA reduction during the late elicitation phase of ACD.

There were some limitations to this study. First, the sample size was small, and a larger sample size is required to provide a more persuasive profile. Second, the study was limited to young women, and only Asian participants were enrolled. Additionally, this study only included Ni and Co as common antigens, and exclusion of other allergens during the study period may have caused bias and inadequacies in the results.

This was the first study to investigate antigen-specific DTH in humans and to study miRNA expression over time during a cell-mediated immunological reaction. Our study provides a valuable starting point to clarify the specific roles of different miRNAs within the positive and negative regulation of inflammatory diseases. Until now, most studies of miRNAs in ACD have used skin biopsy samples. Although the local response can be captured, it is not easy to collect biopsy samples as biomarkers, so we used peripheral blood samples. In our results, let-7d-5p, miR-23b-3p, miR-24-3p, miR-26b-5p, miR-150-5p were significantly downregulated in peripheral blood leukocytes. Some diseases can be diagnosed with high sensitivity and specificity by combining miRNAs [50,51,52]. The ideal biomarker has excellent performance characteristics such as sensitivity, specificity, and positive and negative predictive values. Furthermore, the biomarkers should be relatively safe, easy to measure, and cost-effective. Our study is the first study to investigate antigen-specific DTH in humans and to study miRNA expression over time during a cell-mediated immunological reaction. This report will be a valuable starting point to clarify the specific roles of different miRNAs within inflammatory diseases. In addition, further studies will be conducted using plasma and serum, in which miRNAs are reported to be stable, with the aim of practical application.

## 5. Conclusions

The comparison of miRNA expression profiles between the patient and control groups revealed numerous significant differences. Specifically, five miRNAs (let-7d-5p, miR-23b-3p, miR-24-3p, miR-26b-5p, miR-150-5p) were found to be significantly downregulated in peripheral blood leukocytes on day 7 compared to day 0 in the patient group. These unique miRNA profiles, associated with ACD and inflammation, may prove useful as specific biomarker signatures and therapeutic targets for future investigation.

## Figures and Tables

**Figure 1 biomedicines-11-00277-f001:**
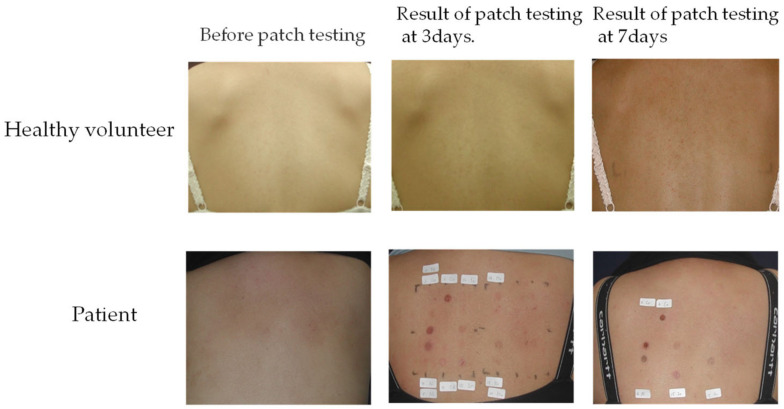
Patch test result. Back photos of the subjects before the patch test, on the 3rd day of the patch test and on the 7th day of the patch test.

**Figure 2 biomedicines-11-00277-f002:**
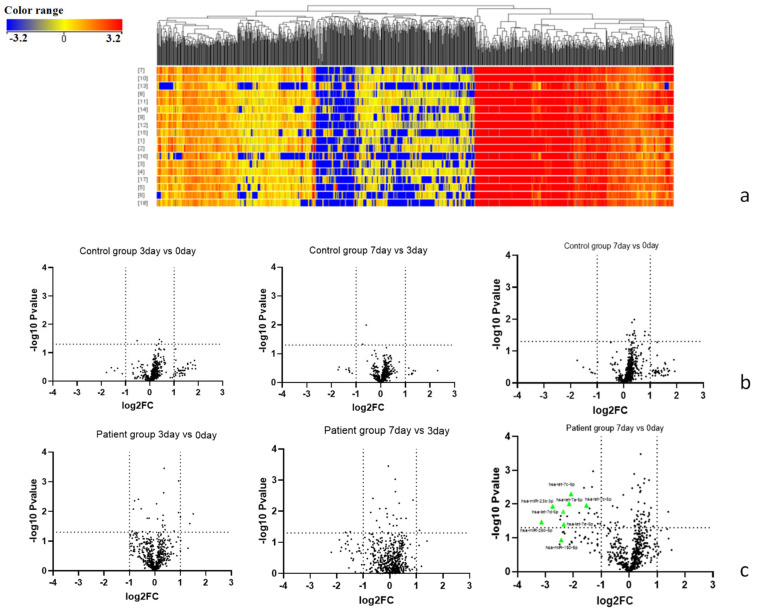
Microarray analysis. (**a**) Heatmap of miRNAs expressed at 3 different time points (day 0, day 3, day 7); (**b**) Volcano plot showing changes in the control group between day 3 and day 0, day 7 and day 3, as well as day 7 and day 0; (**c**) Volcano plot showing changes in the patient group between day 3 and day 0, day 7 and day 3, as well as day 7 and day 0. In the patient group, hsa-let-7 family, hsa-miR-24-3p, hsa-miR-23b-3p, hsa-miR-26b- 5p and hsa-miR-150-5p showed a down-regulation tendency after day 7 compared to day 0 (green triangle).

**Figure 3 biomedicines-11-00277-f003:**
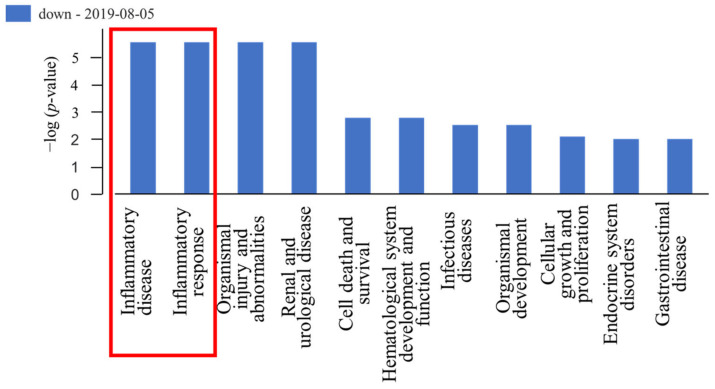
Pathway analysis. Ingenuity pathway analysis showed that the top two diseases related to downregulated miRNAs at day 7 in the patients were inflammatory disease and inflammatory response (red box).

**Figure 4 biomedicines-11-00277-f004:**
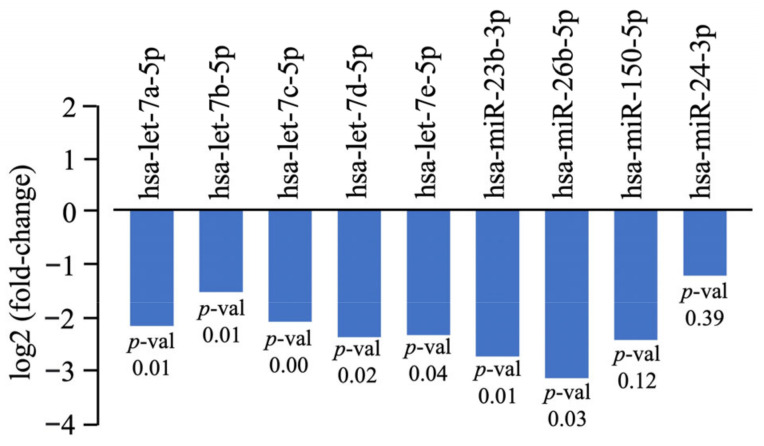
Selected downregulated genes among the patient group at day 7. Bar plot shows a log2 (fold change) value of downregulated genes at day 7 with *p*-value.

**Figure 5 biomedicines-11-00277-f005:**
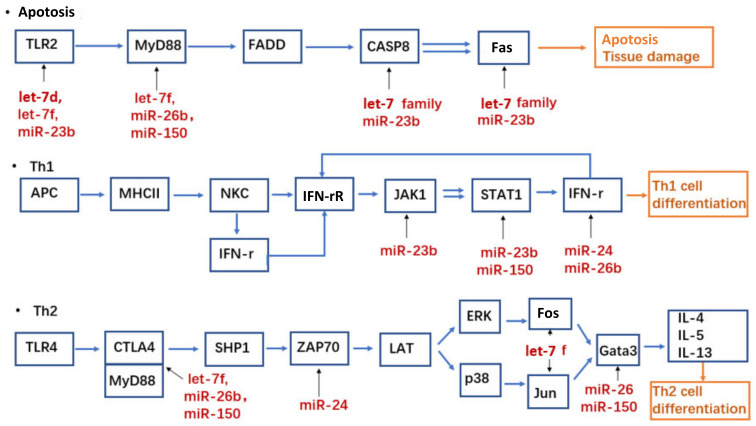
TargetScanHuman database. The prediction of the relations between target miRNAs and mRNAs in the pathways of apoptosis activity, Th1 cell differentiation, and Th2 cell differentiation.

**Figure 6 biomedicines-11-00277-f006:**
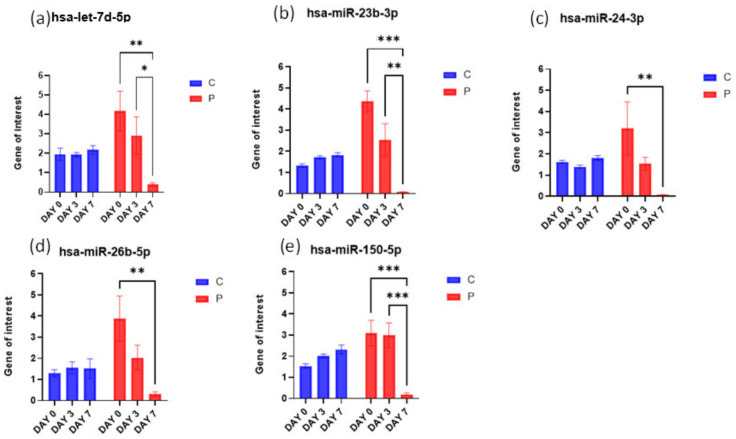
Real-time qPCR results. let-7d-5p, miR-23b-3p, miR-24-3p miR-26b-5p and miR-150-5p expression in peripheral blood leukocytes were compared within the control group (*n* = 3) and patient group (*n* = 3). Expression was normalized to the expression of miR-423-5p. Mean fold change (for each person) in three independent quantitative PCRs is shown. * *p* < 0.05, ** *p* < 0.01, *** *p* < 0.001 (two-tailed *t*-test).

**Figure 7 biomedicines-11-00277-f007:**
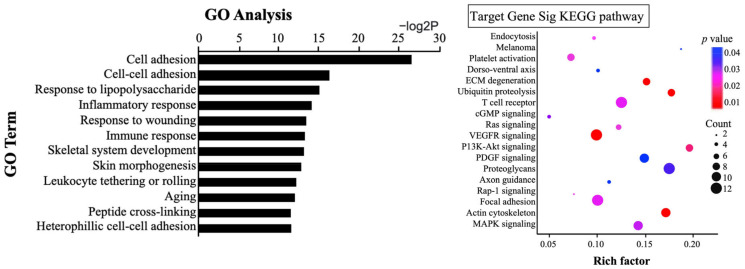
GO analysis and KEGG pathway analysis. Go analysis showed the top 12 miRNA functions. KEGG pathway analysis showed the top 18 pathways regulated by target miRNAs. (*p* < 0.05).

**Table 1 biomedicines-11-00277-t001:** Subjects information (*n* = 6). Demographics of all subjects included in this study.

Subjects	Gender	Age	Race
C1	F	22	Asian
C2	F	26	Asian
C3	F	28	Asian
P1	F	23	Asian
P2	F	23	Asian
P3	F	24	Asian

**Table 2 biomedicines-11-00277-t002:** Allergen type and level of allergy 7 days after the patch test. Clinical scoring of inflammatory reactions induced by metals in all subjects. (*n* = 6).

No	Allergen	%	Vehicle	P1	P2	P3	C1	C2	C3
1	CuSO_4_	2	Aq	**-**	**-**	**-**	**-**	**-**	**-**
2	PdCl_2_	1	Aq	**-**	**+**	**++**	**-**	**-**	**-**
3	K_2_Cr_2_O_7_	0.5	Aq	**++**	**-**	**-**	**-**	**-**	**-**
4	NiSO_4_	5	Aq	**++**	**++**	**+++**	**-**	**-**	**-**
* 5	NiSO_4_	2	Aq	**?+**	**++**	**+++**	**-**	**-**	**-**
6	CoCl_2_	2	Aq	**++**	**+**	**++**	**-**	**-**	**-**
* 7	HgCl_2_	0.1	Aq	**-**	**?+**	**-**	**-**	**-**	**-**
8	HgCl_2_	0.05	Aq	**-**	**-**	**-**	**-**	**-**	**-**
9	SnCl_4_	1	Aq	**-**	**-**	**?+**	**-**	**-**	**-**
* 10	CdSO_4_	1	Aq	**-**	**+**	**-**	**-**	**-**	**-**
11	HAuCl_4_	0.2	Aq	**-**	**-**	**?+**	**-**	**-**	**-**
12	H_2_PtCl_6_	0.5	Aq	**-**	**-**	**-**	**-**	**-**	**-**
13	FeCl_3_	2	Aq	**-**	**-**	**-**	**-**	**-**	**-**
14	InCl_3_	1	Aq	**-**	**-**	**-**	**-**	**-**	**-**
15	IrCl_4_	1	Aq	**++**	**-**	**-**	**-**	**-**	**-**
* 16	MoCl_5_	1	Aq	**-**	**-**	**-**	**-**	**-**	**-**
17	AgBr	2	Pet	**-**	**-**	**-**	**-**	**-**	**-**
18	SbCl_3_	1	Pet	**-**	**+**	**-**	**-**	**-**	**-**
19	ZnCl_2_	2	Pet	**+**	**-**	**-**	**-**	**-**	**-**
20	MnCl_2_	2	Pet	**-**	**-**	**-**	**-**	**-**	**-**
21	CrSO_4_	2	Aq	**-**	**-**	**-**	**-**	**-**	**-**
22	Al_2_O_3_	2	Aq	**-**	**-**	**-**	**-**	**-**	**-**
** 23	TiO_2_	0.1	Pet	**-**	**-**	**-**	**-**	**-**	**-**
** 24	Ti	1	Pet	**-**	**-**	**-**	**-**	**-**	**-**
* 25	BaCl_2_	0.5	Aq	**-**	**-**	**-**	**-**	**-**	**-**
* 26	BaCl_2_	0.1	Aq	**-**	**-**	**-**	**-**	**-**	**-**
* 27	TiCl_4_	0.1	Aq	**-**	**-**	**-**	**-**	**-**	**-**
* 28	TiCl_4_	0.05	Aq	**-**	**-**	**-**	**-**	**-**	**-**

aq: Purified water; pet: Petrolatum; no mark: Patch-test reagents (Torii Pharmaceutical Corp., Tokyo, Japan). **: allergEAZE (SmartPractice Canada Corp., Calgary, AB, Canada). *: Custom-made reagents.

**Table 3 biomedicines-11-00277-t003:** Real-time qPCR primers. Primer details of target miRNAs for real-time qPCR.

miRNA ID	3D-Gene v.16 Microarray Id	Accession *	miRNA Mature Sequence	Primer Sequence
hsa-let-7d-5p	hsa-let-7d	MI0000065	CCUAGGAAGAGGUAGUAGGU	AGAGGUAGUAGGUUGCAUAGUU
hsa-miR-23b-3p	hsa-mir-23b	MI0000439	CUCAGGUGCUCUGGCUGCUU	AUCACAUUGCCAGGGAUUACCAC
hsa-miR-26b-5p	hsa-mir-26b	MI0000084	CCGGGACCCAGUUCAAGUAA	UUCAAGUAAUUCAGGAUAGGU
hsa-miR-150-5p	hsa-mir-150	MI0000479	CUCCCCAUGGCCCUGUCUCCC	UCUCCCAACCCUUGUACCAGUG
hsa-miR-24-3p	hsa-miR-24-3p	MIMAT0000080	UGGCUCAGUUCAGCAGGAACAG	TGGCTCAGTTCAGCAGGAAC

* Accession in miRbase database http://www.mirbase.org/ (accessed on 11 November 2019).

## Data Availability

miRbase database http://www.mirbase.org/ (accessed on 11 November 2019). TargetScanHuman http://www.targetscan.org/vert_72/ (accessed on 19 October 2019). GeneSpring http://www.ncbi.nlm.nih.gov/projects/geo/ (accessed on 13 August 2019).

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
