# Peer review of "Identification of microRNA Signatures in Peripheral Blood of Young Women as Potential Biomarkers for Metal Allergy"

_biomedicines, 2023, doi:10.3390/biomedicines11020277_

Round 1
Reviewer 1 Report
Yuehui et al. investigated the role of microRNAs in metal allergies and obtained valuable results. However, several things should be improved.
In the abstract, the authors should include the number of study participants.Lines 45-48 - please provide the bibliographic source.
Lines 56-57 – please provide more details about the pathways and networks involved
Lines 71-72 - please provide the bibliographic source.
In the Introduction Section, the authors specify that metal allergies are more common in women than in men. However, when the authors define the aim and objectives of the research, it is necessary to clearly explain why only women were included in the study.
The participants in the patient group showed signs of Ni–Co allergic dermatitis. The authors included in the study only patients with Ni-Co allergic dermatitis and excluded patients with allergic dermatitis to other metals? This fact should be specified in the Material and Method Section.Lines 310-313 - please provide the bibliographic source
Line 338 – "certain cytokines...". please provide the cytokines that were reported to be involved.
In the Discussion section, the authors should include more details about the implications of the results obtained in the diagnosis and management of these patients. The conclusions are briefly presented. The authors should improve this section and present the conclusions of the study more comprehensively.Author Response
Please see the attachment.

Reviewer 2 Report
The manuscript is interesting. The Authors have evaluated miRNA expression in cases of metal allergy confirmed by patch test. The results are well described in the text of the manuscript and adequately presented in tables and figures. An important limitation of the study is the low number of participants - 3 participants each within a group. However, the Authors are aware of this problem, as they mention this limitation in the manuscript.
In my opinion, the manuscript requires the following corrections:
1. Table 3 should be graphically revised.
2. In my opinion, the conclusions are too general and contain too little information about the studies conducted, which means that they do not represent the authors' observations made during the studies.
Round 2
Reviewer 1 Report
The manuscript has been significantly improved
it could be published in current form